# Relationship between Attention Capacity and Hand–Eye Reaction Time in Adolescents between 15 and 18 Years of Age

**DOI:** 10.3390/ijerph191710888

**Published:** 2022-09-01

**Authors:** Álvaro Huerta Ojeda, Patricio Lizama Tapia, Jaime Pulgar Álvarez, Claudia González-Cruz, María-Mercedes Yeomans-Cabrera, Juan Contreras Vera

**Affiliations:** 1Núcleo de Investigación en Salud, Actividad Física y Deporte ISAFYD, Universidad de Las Américas, Viña del Mar 2531098, Chile; 2Facultad de Educación, Universidad de Las Américas, Viña del Mar 2531098, Chile; 3Facultad de Educación, Universidad Andres Bello, Santiago 8370035, Chile

**Keywords:** executive functions, attention, concentration, reaction capacity, adolescents

## Abstract

Some experiences demonstrate a direct correlation between attention capacity and reaction capacity. However, the evidence from adolescents enrolled in the Chilean school system is scarce. The purpose of this study was to determine the relationship between attention capacity and hand–eye reaction time (RT) in adolescents between 15 and 18 years of age. Thirty-one adolescents participated voluntarily in this study. The variables were: attention capacity, evaluated through the *Evalúa-10* battery (item 1.1), and hand–eye RT, assessed through a simple RT test (SRT) and complex RT (CRT). The relationship between the variables was performed through Pearson’s correlation. Comparisons between males and females were performed with the *t*-test for independent samples (*p* ˂ 0.05). There was a moderate correlation between attention and CRT (*r* = −0.43), a very high correlation between attention and SRT in males (*r* = −0.73), and between attention and CRT in females (*r* = −0.73). Between males and females, there was no difference in attention (*p* ˃ 0.05), while males showed better RT in all tests (*p* ˂ 0.05). Attention positively influences hand–eye RT in both males and females. Likewise, male adolescents present better hand–eye RT than their female peers.

## 1. Introduction

Good brain health is defined as a dynamic state of cognitive, emotional, and motor domains throughout life supported by physiological processes; it is multidimensional and can be measured objectively and experienced subjectively [1]. In this context, different initiatives worldwide exist to create a map of neural networks in humans. Although these projects are still under development, they have enabled progress in understanding the brain [2]. In this sense, within the executive role of the brain, three functional units are distinguished: (1) limbic and reticular systems—both in charge of excitation and motivation, (2) postrolandic cortical areas—which receive, process, and store information, and (3) frontal lobes—which program, control and verify activities [3]. Additionally, the observation of the frontal lobes has allowed us to verify that some of the intellectual behaviors (such as planning, self-control, problem-solving, reasoning, and working memory) are controlled by these areas of the brain; these are fully defined as the executive functions (EF) [3].

Specifically, EFs comprise those skills that allow human beings to set goals and carry them out through planning and monitoring by inhibiting thoughts, behaviors, and emotions that interfere with their achievement [4]. This leads to more flexible and autonomous goal-oriented behaviors [5]. In anatomical terms, the EFs are integrated by the prefrontal cortex (PFC); this brain area is considered an area of integration of the prefrontal cortex [6]. Likewise, attention capacity is within the EF [7], although some researchers consider it an independent and/or a higher psychological function [8]. Attentional capacity is defined as the mind taking clear and vivid possession of only one of several simultaneous objects or sequences of thought [9]. Thus, attention is the voluntary and involuntary behavior adopted by the individual, through which specific contents constitute the center of their conscious life [10]. Attentional capacity is regulated by the medial prefrontal cortex (MPFC) [11]. MPFC is indispensable both for the creation of new neural connections and for the formation of stable brain circuits [12].

Specifically, attention capacity is necessary, as the brain receives more information than can be processed at any given time [10]. In this context, three specific purposes have been described for the capacity of attention: (1) it favors precision in the perception of objects and the correct execution of particular actions—especially when other objects or other actions are present as an option; (2) it decreases the time of perception and action to prepare the system that processes them; and (3) it helps to maintain concentration or action for the necessary time [8]. Furthermore, within the capacity of attention is selective or focal attention. This type of attention is directed to processes that respond to specific stimuli or tasks while ignoring other available stimuli or tasks [13]. In this sense, selective or focal attention has gained significant relevance due to the changing environment surrounding the human being and the brain’s limited capacity to process information at every moment [12]. Therefore, concentration or wakefulness (localization of sustained attention on the relevant aspects of a task) directs the totality of awareness—and not only a part of it—to a specific task for a prolonged period, quickly or slowly [13].

On the other hand, reaction time (RT) is a physical capacity that seems closely related to attention capacity (RT) [14]. This physical capacity is defined as the time that elapses from the moment we perceive a stimulus until we make a response as a consequence of that stimulus [15]. In turn, the neurological factors that influence this RT are (1) a receptor organ (kinesthetic, visual, or auditory), (2) the length of the sensory pathway, and (3) the type of axon or number of synapses from signal reception to the receiving cerebral cortex [16]. Likewise, RT varies depending on factors such as the type and complexity of the stimulus, the state of the organism, and the sensory modality stimulated [15]. In effect, the motor response is transmitted by efferent neuronal pathways passing through the spinal cord and responding to the motor units that implement the desired motor action [17].

As described above, there is evidence of an association between attention capacity and RT and the relationship between attention capacity and RT [14]. Specifically, it has been observed that the dorsolateral region of the frontal lobe receives highly processed sensory information from cortical association areas located in the parietal, occipital (vision), and temporal cortexes [18]. In turn, the dorsolateral cortexes send information to the basal ganglia’s (cognitive) regions, premotor cortexes, and sensory-association cortexes [18]. In this regard, evidence shows that specific RT, such as visual–motor reaction time, is highly dependent on the speed of perception and processing in the brain’s visual movement system when responding to optical signals [19]. It has also been shown that the connections from the frontal cortex to more caudal cortical areas are responsible for implementing attention mechanisms [18]. Therefore, a low attention capacity could affect the focusing, selection, and blocking of unnecessary information and, consequently, the RT executing tasks [17]. To our knowledge, no studies relate these variables to Chilean adolescents. Therefore, the primary purpose of this study was to determine the relationship between attention capacity and hand–eye RT in adolescents between 15 and 18 years of age. The secondary objective was to determine the relationship by sex between the variables. This study hypothesized that a greater attention capacity would generate a shorter hand–eye RT in adolescents.

## 2. Materials and Methods

### 2.1. Participants

The sample size was calculated with a statistical program (G*Power, v3.1.9.7, Heinrich-Heine-Universität, Düsseldorf, Germany). The combination of tests used in the statistical program to calculate the sample size was as follows: (a) exact, (b) correlation: bivariate normal model, and (c) a priori: compute required size—given α, power, and effect size. Tests considered two tails, r H1 = 0.48, α-error < 0.05 and a desired power (1-β error) = 0.8, slope r H0 = 0.00, the minimal sample size was 31 participants [20].

Thirty-one adolescents (19 males and 12 females) between 15 and 18 years of age from educational establishments in the Greater Valparaíso voluntarily participated in this study. The inclusion criteria were to be male or female, between 15 and 18 years of age, and studying in an educational establishment in Greater Valparaíso. In contrast, the exclusion criteria were: to be a competitive athlete, have the physical impossibility to perform some of the proposed tests, refuse to sign the informed consent form by parents or guardians, and the presence of attention deficit disorder declared by the participants. All participants were informed of the objectives of the study. Before applying the protocols, all participants signed an informed assent, while their parents or guardians signed informed consent. The Universidad Central Ethics Committee approved the study, informed assent, and informed consent (registration number: 54/2022). In addition, the study was developed under the ethical standards for exercise and sports sciences [21].

### 2.2. Research Design

Observational research with an associative and cross-sectional strategy. Associative, since it explores the relationships between variables to predict or explain their behavior [22] (Figure 1).

### 2.3. Anthropometric Measurements

For the characterization of the sample, weight, height, body mass index (BMI), and body fat percentage were evaluated. Height (cm) was assessed using a stadiometer from the feet to the vertex (Frankford plane). Weight (kg) and body fat percentage were assessed using a Tanita Inner Scan BC-554 digital scale. Weight, height, and fat percentage were assessed with the students barefoot, wearing shorts and a light T-shirt. BMI was interpreted according to anthropometric standards to assess nutritional status [23].

### 2.4. Psychopedagogical Battery Evalúa-10

The Psychopedagogical Evaluation Battery *Evalúa-10* (E-10) aims to measure general and specific cognitive abilities and adaptation levels [24]. The results of these tests allow decisions to be made regarding the teaching–learning processes. The E-10 battery consists of 7 items: (1) cognitive scale, (2) reading scale, (3) writing scale, (4) mathematics scale, (5) psychosocial scale, (6) vocational scale, and (7) study strategy scale [24]. However, for the development of this research and associated with the purpose of the study, only the first section of the cognitive scale was used (1.1: attention-concentration). Based on visual stimuli, the first section of the cognitive scale evaluates the ability to maintain concentrated attention (1.1). This section consists of two tasks: Task 1, “Identification of the errors made in a number-drawing association”; this task presents 68 figures, while the correction awards one point for a correct guess (A) and subtracts one point for an error (E), considering omissions (O) as an error. If the participant does not complete task 1, the answers that should not have been marked are added as A, and the answers that should have been marked as E. Score Task 1 = Ʃ A − (E + O). Task 2, “Quick location of drawings identical to a given model requiring analytical observation and selective attention”; this task presents 98 figures. However, only 97 score (the first figure in Task 2 is the example for explanation, therefore, it was not considered in the scoring). The correction awards one point for A and subtracts one point for E, considering the Os as an error; in the case that the participant does not complete Task 2, the answers that should not have been marked are added as A, and the answers that should have been marked are added as E. Task 2 score = Ʃ A − (E + O). To obtain the score for item 1.1 (attention-concentration) of E-10, the scores for Task 1 and Task 2 are added together: Score 1.1 = Ʃ Task 1 + Task 2. The latter score was entered for statistical analysis [24].

### 2.5. Reaction Time

The purpose of this test is to measure the time it takes participants to turn off a series of visual stimuli in milliseconds (ms). The BlazePod© wireless LED light device, Version 2.6.5 (Canada), was used to measure the hand–eye RT. This wireless device is composed of six sensors. These sensors were placed on an 80 cm (cm) high table and distributed in a semicircle. The arrangement of the devices was 40 cm from the center of the table and separated by 20 cm between them [14]. To perform the hand–eye RT test, each participant sat in front of the devices, with their hands in contact with the edge of the table (Figure 2). Three RT tests were performed (two simple and one complex RT). The simple RT test (SRT) was performed first with the right hand (48 blue stimuli) and then with the left hand (48 red stimuli). Each device was randomly turned on during testing eight times, maintaining the order for all participants. The complex RT test (CRT) was performed with both hands and considered 48 stimuli (eight stimuli per device). For the CRT test, each device was turned on four times with blue and four times with red lights (eight times for each device). If the light was turned on blue, the participant had to turn it off with the right hand, and if it was turned on with red light, the participant had to turn it off with the left hand. In this test, the switching on of the lights in each device was randomized. However, the switching sequence was the same for all participants in the study. This test’s error was not considered in the statistical analysis (e.g., turning off lights with the wrong hand). The separation between the three RT tests was 60 s (s) [14].

### 2.6. Data Analysis

The E-10 test, RT, and anthropometric data were arranged in a spreadsheet designed for the study. Descriptive data are presented as means and standard deviations (SD). The normal distribution of the data was confirmed by the Shapiro–Wilk test (*p* > 0.05). Pearson’s test calculated the correlation between the variables [25]. The criteria for interpreting the strength of the r coefficients were as follows: trivial (<0.1), small (0.1–0.3), moderate (0.3–0.5), high (0.5–0.7), very high (0.7–0.9), or practically perfect (>0.9). Differences between males and females were calculated using the *t*-test for independent samples [26]. The effect size (ES) for the *t*-test was calculated with Cohen’s *d* test. The latter analysis considers an insignificant (*d* < 0.2), small (*d* = 0.2 to 0.6), moderate (*d* = 0.6 to 1.2), large (*d* = 1.2 to 2.0) or very large (*d* > 2.0) effect. All statistical analyses were performed with Prism version 7.00 for Windows software. The significance level for all statistical analyses was *p* < 0.05.

## 3. Results

The mean age of the adolescents evaluated was 16.4 ± 0.99 years. This analysis found no significant differences between males and females for age and anthropometric parameters (*p* > 0.05) (Table 1).

At the end of the analysis, the 31 participants achieved 99.1 ± 24.8 points for section 1.1 (attention-concentration) of the E-10 test. In this test, no differences between males and females were evident (*p* ˃ 0.05). In parallel, the SRT was 558.9 ± 70.6 ms for the right hand and 570.4 ± 72.5 ms for the left hand. The CRT was 713.2 ± 85.9 ms for both hands. In the three tests used to measure the TR, differences between males and females were evident (*p* ˂ 0.05) (Table 2).

When analyzing the concordance between attention-concentration ability and CRT for the 31 adolescents, a moderate correlation was observed (*r* = −0.43, *p* ˂ 0.05). The same analysis evidenced a high correlation in males (*r* = −0.61, *p* ˂ 0.05) and very high correlation in females (*r* = −0.73, *p* ˂ 0.05). The correlation analysis is reported in Figure 3.

## 4. Discussion

The present study aimed to determine the relationship between attention capacity and hand–eye RT in adolescents between 15 and 18 years of age. Likewise, this research related the study variables by sex. The results showed a moderate relationship between attentional capacity and hand–eye RT for all participants. However, when connecting attention-concentration capacity to SRT and CRT by sex, high and very high concordances were observed for both males and females.

### 4.1. Attention and Concentration

Regarding attention capacity, after applying section 2.4 of E-10, no significant differences were evident between males and females (*p* ˃ 0.05). However, females had 10.47 average points over males in the same test. In this sense, Liu et al. [27], after studying attentional networks and their relationship with spatial orientation function in young adults between 18 and 26 years of age (22.6 ± 1.3 years), observed a better functioning of this specific area in female. Likewise, Sebastián et al. [28] studied attentional processes in military tasks executed by males (31.8 ± 14.1 years) and females (30.5 ± 10.7 years), concluding that females present greater brain activity and are more selective than males since they only activate the brain area they require for the specific task. In contrast, males activate almost their entire brain to perform the same task. Gutiérrez-Ruiz et al. [29] studied sex differences in a component of executive control (ability to switch tasks) in undergraduate students (24 females: 19.7 ± 1.8 years and 24 males: 19.8 ± 1.7 years) after studying sex differences; no significant differences between genders were found, demonstrating that, when the level of complexity increases, the success rate decreases and the response time increases in both sexes. In the present study, the nonsignificant differences observed in attention span between men and women (*p* ˃ 0.05) suggest that they are not inherent to sex but to other variables that might covary with it and that physical condition or even the phase of the menstrual cycle might influence attention [30]. Likewise, with item 1.1, the E-10 is a valid test to assess attention-concentration ability [24]. However, to our knowledge, there are no public reports evidencing the level of this skill in Chilean students, and even fewer that relate attention-concentration ability with hand–eye RT in Chilean adolescents. Therefore, the present study is pioneering in the area, but it is necessary to continue exploring this topic.

### 4.2. Reaction Time

Concerning hand–eye RT, our findings showed that males responded better than females to visual stimuli in all the tests applied (*p* ˂ 0.05). In this regard, Neto et al. [31] reported significant differences in RT values between males and females (males: 243 ± 67 ms, females: 342 ± 50 ms, *p* = 0.011), whereas, after a hand–eye RT test, Reigal et al. [14] reported an average difference of 46.9 ms between males and females. Furthermore, Szabo et al. [32] reported significant differences after applying the Ruler Drop Test, which assesses hand–eye reaction speed when comparing males and females in the dominant hand (*p* = 0.095). In this study, which evaluated males and females aged 14–15 years, no significant differences were found between genders when comparing reaction speed in the non-dominant hand (*p* = 0.095). [32]. Contrary to this evidence, Vences de Brito et al. [17] found evidence that female Shotokan karateka had better RT than males practicing the same sport (male: 292 ± 30 ms, female: 288 ± 20 ms, *p* = 0.05). The differences in hand–eye RT likely observed in the different studies may be conditioned to different modulations in neural activity in the visual and motor regions of the cerebral cortex [19]. Likewise, it has been observed that men and women who practice sports integrate sensory information better in executing motor actions based on reaction speed and coordination [32]. In this sense, Akarsu et al. [33] compared sport-related visual abilities such as the hand–eye reaction time of athletes with non-athletes. They showed that male and female athletes generate better hand–eye RTs than non-athletes. Therefore, although more studies are needed to establish the causes of the differences between males and females, it is advisable to include more sports activities in schools to develop this and other physical abilities optimally.

Additionally, evidence has been found that hand–eye RT improves over the years [34], provided that there is an adequate training process [14,17]. In this sense, Reigal et al. [14] reported an improvement of 6.7 and 4.0% in hand–eye RT for the ranges between 10–11 and 11–12 years of age, respectively. In parallel, Vences de Brito et al. [17] provide evidence of a better SRT in an older age group (15–19 years: 294 ± 26 ms; 20–35 years: 293 ± 24 ms and +35 years: 290 ± 31 ms). The response to this increase in reaction capacity over the years would be justified, among other factors, by the maturation of the central nervous system since this system determines the speed of visuomotor processes [35]. Another factor that would justify an increase in reaction capacity is increased brain activity [34]. Indeed, Adleman et al. [34] related brain activity to reaction capacity, concluding that (a) higher brain activity generates lower RT and (b) this relationship increases with age, provided that there are no associated disorders in cognitive functioning [36]. Consequently, the relationship between brain activity and responsiveness may reflect significant changes throughout development.

### 4.3. Physical Exercise and Brain Activation

The findings in the present study suggest that greater attention-concentration ability improves both SRT and CRT in adolescents between 15 and 18 years of age (SRT: *r* = −0.43, CRT: *r* = −0.39). At the same time, the analysis by sex evidenced high and very high concordances between the abilities assessed for both males and females. In parallel, it has been shown that physical exercise improves reaction capacity and, therefore, attention and concentration [14,37]. In addition, it has been observed that children with obesity, between 6 and 12 years of age, have a higher RT than their normal-weight peers [38]. In this regard, childhood obesity has been associated with impaired perceptive-motor function; however, a better understanding of the mechanisms underlying motor incompetence in children with obesity is needed [38]. Furthermore, Reigal et al. [14] analyzed the relationship between RT, selective attention, concentration, and physical fitness in children between 10 and 12 years old, concluding that better development of selective attention and concentration, as well as an increase in physical fitness, improve RT at these ages. Budde et al. [37] observed the influence of coordination exercises, in short, training sessions, on cognitive performance in healthy adolescents aged 13 to 16 years, concluding that coordination exercises generate a prior activation of parts of the brain, an area responsible for mediating functions such as attention. Specifically, to increase cognitive performance in adolescents, Budde et al. [37] suggested including coordination exercises in short training sessions during the school day. Added to the above mentioned, there is evidence that a correct psychomotor stimulation concordant with age would help improve efficiency in daily tasks, assisting children’s and adolescents’ personal and social growth [14]. However, the brain mechanisms that relate both white matter plasticity and cortical plasticity to changes in RT resulting from training have not yet been fully understood. They, therefore, need further exploration [39]. Therefore, taking into account that physically active people with better physical fitness show better hand–eye RT in both males and females [14,32,33], and that coordination exercises influence the activation of parts of the brain (a situation that could trigger better cognitive performance) [37], it is advisable to include more sports activities in all phases of child and adolescent development, especially in educational centers.

### 4.4. Limitations

The pandemic generated by COVID-19 between 2020–2022 has led to a decrease in face-to-face activities in educational establishments. This situation, together with the distrust experienced by parents and/or guardians of physical contact with adolescents, led to a low acceptance of participation in the study. This situation (low number of participants) did not allow us to compare different ages and educational establishments or to observe the effect of anthropometric variables (BMI) on attention-concentration capacity or RT.

## 5. Conclusions

Attention has a positive influence on the hand–eye RT in both males and females. Likewise, male adolescents present better hand–eye RT than their female peers. These findings could help identify potential targets and generate specific interventions in adolescents.

## Figures and Tables

**Figure 1 ijerph-19-10888-f001:**
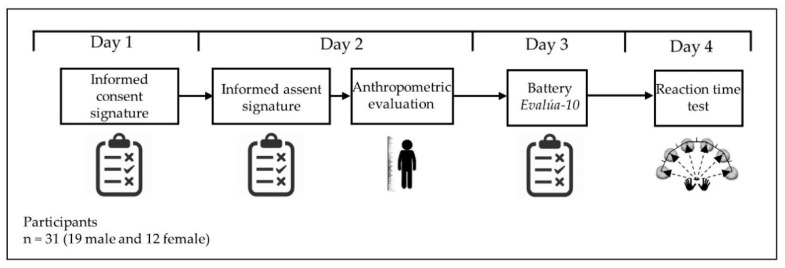
Research design.

**Figure 2 ijerph-19-10888-f002:**
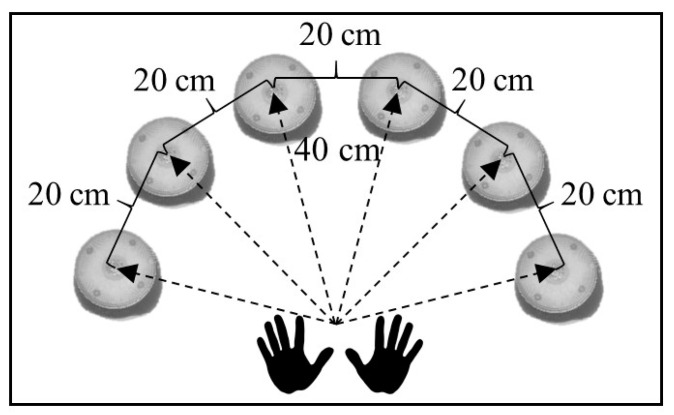
Distribution of devices in the reaction time test.

**Figure 3 ijerph-19-10888-f003:**
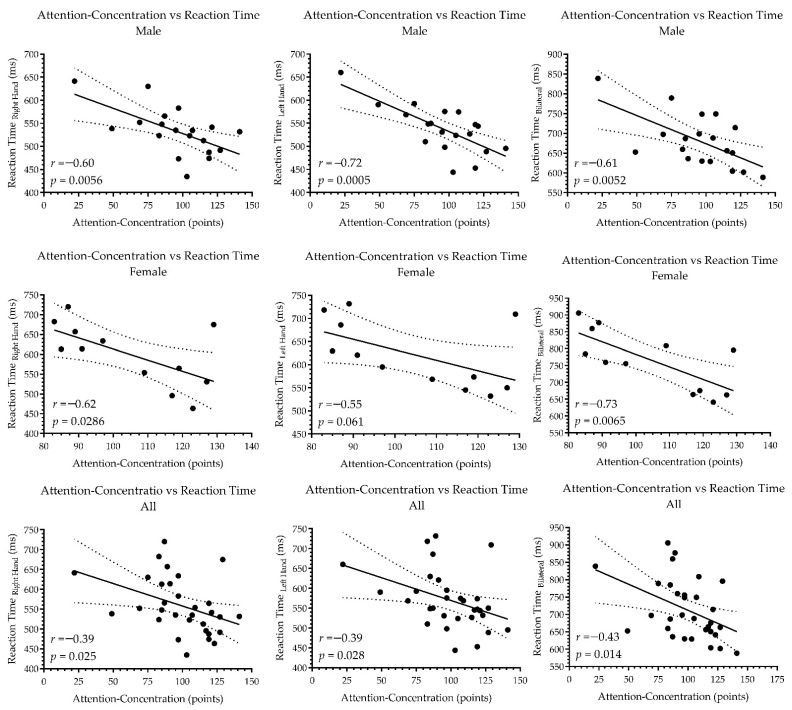
Correlation analysis between the capacity of attention-concentration and reaction time.

**Table 1 ijerph-19-10888-t001:** Sample characterization.

	All (*n* = 31)	Male (*n* = 19)	Female (*n* = 12)	Mean Diff	95% CI of Diff	t	ES	*p*-Value
Age (years)	16.4 ± 0.99	16.2 ± 1.08	16.7 ± 0.78	−0.45	−46.04 to 45.13	0.029	0.49	ns
Weight (kg)	69.1 ± 14.5	71.4 ± 16.6	65.4 ± 9.86	6.02	−39.57 to 51.61	0.391	0.45	ns
Height (m)	1.65 ± 0.10	1.71 ± 0.08	1.55 ± 0.05	0.15	−45.44 to 45.74	0.009	2.41	ns
BMI (kg/m^2^)	25.4 ± 4.45	24.4 ± 4.75	27.0 ± 3.56	−2.56	−48.16 to 43.02	0.166	0.62	ns
Fat (%)	25.4 ± 9.35	20.6 ± 7.96	33.2 ± 5.31	−12.62	−58.21 to 32.97	0.820	1.90	ns
DH	R = 26 (83.8%) L = 5 (16.2%)	R = 15 (78.9%) L = 4 (21.1%)	R = 11 (91.6%) L = 1 (8.4%)	-	-	-	-	-
Non-DH	R = 5 (16.2%) L = 26 (83.8%)	R = 4 (21.1%) L = 15 (78.9%)	R = 1 (8.4%) L = 11 (91.6%)	-	-	-	-	-

CI, confidence intervals; DH, dominant hand; diff, difference; ES, effect size; kg, kilograms; kg/m^2^, kilograms per meters squared; L, left; m, meters; Non-DH, non-dominant hand; ns, not significant; R, right.

**Table 2 ijerph-19-10888-t002:** Comparison of attention-concentration capacity and reaction time by sex.

	All (*n* = 31)	Male (*n* = 19)	Female (*n* = 12)	Mean Diff	95% CI of Diff	t	ES	*p*-Value
E-10 (points)	99.1 ± 24.8	95.6 ± 28.4	104.7 ± 17.7	−10.47	−54.68 to 36.50	0.590	0.39	ns
Reaction Time Right hand (ms)	558.9 ± 70.6	532.7 ± 50.9	600.3 ± 79.4	−67.18	−113.2 to −21.99	4.392	1.04	0.0002
Reaction Time Left hand (ms)	570.4 ± 72.5	538.0 ± 51.6	621.8 ± 72.7	−77.37	−129.4 to −38.20	5.446	1.35	0.0001
Reaction Time Bilateral (ms)	713.2 ± 85.9	679.9 ± 66.0	766.0 ± 89.8	−77.02	−131.6 to −40.45	5.592	1.10	0.0001

CI, confidence intervals; diff, difference; ES, effect size; E-10, *Evalúa-10* test (attention-concentration), ms, milliseconds; ns, not significant.

## Data Availability

The database of the study can be downloaded from the following link: https://doi.org/10.6084/m9.figshare.20069378 (accessed on 14 July 2022).

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
