# Peer review of "Relationship between Attention Capacity and Hand–Eye Reaction Time in Adolescents between 15 and 18 Years of Age"

_ijerph, 2022, doi:10.3390/ijerph191710888_

Round 1

Reviewer 1 Report

Comments:

It is of great significance for the researchers to analyse the relationship between attention capacity and hand-eye reaction time in adolescents. The findings of the study may facilitate the potential targets identification and specific interventions generation for adolescents. However, due to the limitation in the sample size and the sample singleness, the vertical comparison between various age groups, and the horizontal comparison in different educational establishments are unavailable, which are in the way of the deep investigation of the study. Also, there are still some details that need to be improved.

Details and Suggestions:

1.      In the Introduction section, the authors mainly described the possible mechanisms or the evidence of the association and relationship between attention capacity and RT. However, readers might also want to know whether the study done by the authors had done before in other population or other age group. Are there any previous studies? And what have they found?

2.      Also in the introduction, the authors hypothesized that “a greater attention capacity would generate a lower hand-eye RT in adolescents”. Maybe it is better to use long/longer or short/shorter to describe time used for hand-eye reaction.

3.      In the Materials and Methods section, apart from the existing statement, did the authors consider other disease/health status that may influence the attention capacity and RT?

4.      In the Materials and Methods, “the figure from 1 this task 2 is the example”, did the authors want to explain that the first figure in task 2 is the example for explanation?

5.      Regarding the Discussion, 4.2 Reaction time, line 278-290, sentences “Additionally, ……, provided that there are no associated disorders in cognitive functioning [33].” have appeared in line 267-278, please avoid duplication.

6.      In Discussion, 4.3Physical exercise and brain activation, how did the authors turn to the statement, “In this sense, it has been shown that physical exercise improves reaction capacity and, therefore, attention and concentration [14,34].”. Before this statement, the authors only said that “the analysis by sex evidenced high and very high concordances between the abilities assessed for both males and females.”, without the statement or demonstration related to physical exercise.

Reviewer 2 Report

Comments to the Author

The manuscript titled "Relationship between Attention Capacity and Hand-Eye Reaction Time in Adolescents Between 15 and 18 Years of Age" analyzes the relationship between attention capacity and hand-eye reaction time in adolescents. However, there are several points that require further clarity;

1- Page 1, Line 13: Do adolescents in the Chilean school system differ from other adolescents around the world? Why did you feel the need to emphasize this? I just want to know if this population is special.

2- Page 1, Line 18-22: Sex and gender are both generally referred to in two distinct categories: male and female or man and woman. First though, it is necessary to point out that the terms sex and gender are not synonyms. Even the terms male and female, man and woman are not interchangeable. Therefore, you should be careful to use the same term throughout the article. My advice is to change men and women to male and female.

3- Page 3, Line 102: Reference should be to the work used as a reference in the power analysis.

4- Page 2, Line 96: It is important to indicate whether there are athletes among the participants. If so, indicate which fields you are interested in.

5- Page 3, Line 130: You should provide more details about the methods of anthropometric measurements.

6- Page 3, Line 136: What exactly is the source of the method used for attention? Is it a book or an article or another? I couldn't find any way to reach it. Also, I think that using a single item (1.1: Attention-Concentration) of a scale consisting of 7 items may cause serious methodological mistakes. Please explain.!

7- Page 3, Line 157: please, add reference

8- Page 3, Line 177: please, add reference

9- Page 5, Line 205: In Table 1, the number of rights and left hands dominant and non-dominant for male and females (n, %) should be given.

10- Page 6, Line 240-254: This section is insufficient so you must improve. It would be better if you share the results of studies using the same method as you (1.1: Attention-Concentration). On the other hand, you must indicate the ages of the male and female in the studies you have mentioned here. Finally, explain why there is no difference in attention between male and female in your study and support it with references.

11- Page 7, Line 267-291: You do not make age comparisons in your work. Why do you feel the need to discuss reaction time by age? This seems so unnecessary. Please focus on why the reaction time differs between male and female and support the results with references.

12- Page 7, Line 292: There is no need for such a section title. Please remove the title.

13- Page 7-8, Lines 293-316: There is no need for such a section title. Please remove the title. Also in this section you need to discuss the relationship between attention-concentration and reaction time. Perhaps you could mention the effects of exercise on reaction or coordination at the end of this line. Please revise deeply and focus on your main topic.

GENERAL COMMENTS:

1. The manuscript requires language improvement.

2. The topic is important but especially the discussion section should be improved

significantly. Literature review is nonadequacy.

3. Abstract should be re-edited after changes made in the article.

4. Some abbreviations are unnecessary (Attention Deficit Disorder (ADD)).

Round 2

Reviewer 1 Report

The authors well revised the manuscript and I have no other questions.

Reviewer 2 Report

congratulations